# Revisiting Populations in Multi-Agent Communication

## Abstract

Despite evidence from sociolinguistics that larger groups of speakers tend to develop more structured languages, the use of populations has failed to yield significant benefits in emergent multi-agent communication. In this paper we reassess the validity of the standard training protocol and illustrate its limitations. Specifically, we analyze population-level communication at the equilibrium in sender-receiver Lewis games. We find that receivers co-adapt to senders they are interacting with, which limits the effect of the population. Informed by this analysis, we propose an alternative training protocol based on "partitioning" agents. Partitioning isolates sender-receiver pairs, limits co-adaptation, and results in a new global optimization objective where agents maximize (1) their respective "internal" communication accuracy and (2) their alignment with other agents. In experiments, we find that agents trained in partitioned populations are able to communicate successfully with new agents which they have never interacted with and tend to develop a shared language. Moreover, we observe that larger populations develop languages that are more compositional. Our findings suggest that scaling up to populations in multi-agent communication *can be* beneficial, but that it matters *how* we scale up.

## 1 Introduction

Uncovering the mechanisms that underlie our ability to communicate using language is an important stepping stone towards developing machine learning models that are capable of coordinating and interacting via natural language. Over the last few years, there has been increasing interest in simulating the emergence of language using artificial agents trained with reinforcement learning to communicate to achieve a cooperative task [33]. Typically, agents are trained to perform a variant of the Lewis signaling game [37, 51] wherein a *sender* emits a message describing an object and a *receiver* attempts to reconstruct the object based on the description. This line of work has applications to semi-supervised learning applied to concrete tasks such as image captioning or representation learning [36, 18].

Most previous research has focused on communication between a single pair of agents. However, there is mounting evidence that the communication protocols developed in this restricted setting become highly specialized and exhibit properties that are at odds with those found in human languages [4, 8]: for example agents are able to solve the task successfully while using languages that are not compositional [32, 9]. As a possible solution, a growing body of work is advocating for scaling up the emergent communication literature to populations of more than two agents communicating simultaneously [24, 30, 49, 10]. Indeed, there is substantial evidence within the language sciences that population dynamics shape the language structure [47, 42]. In spite of this fact, several negative results have been obtained, showing that training agents in population yield marginal benefits without explicit pressure towards *e.g.* population diversity [49] or emulation mechanisms [10].

Submitted to 36th Conference on Neural Information Processing Systems (NeurIPS 2022). Do not distribute.

In this paper, we call into question the way such populations are trained. By studying a simple referential game, we evaluate populations on two desirable features observed in natural language:

- Agents are able to communicate with new partners within the same population [23]

- Larger populations tend to develop more structured languages [42].

We provide evidence that populations of artificial agents do not always possess these features (as also attested by previous work, *e.g.* Kim and Oh [30], Chaabouni et al. [10]). To shed light on this phenomenon, we analyze the behaviour of agents in a population at the equilibrium. We find that with the standard training procedure, the functional form of the objective is the same as that of a single pair of agents, due to receivers co-adapting to their training partners. As our main contribution, we propose an alternative training procedure which *partitions* sender-receiver pairs and limits co-adaptation of receiver agents. We show that this new training paradigm maximizes a different objective at the population level. In particular, it explicitly promotes mutual-intelligibility across different agents.

In experiments, we find that agents trained in partitioned populations are able to communicate successfully with new communication partners with which they have never interacted during training, and that languages spoken by various agents tend to be similar to one another. In addition, we observe that (1) languages developed in partitioned populations tend to be more compositional and (2) there is a population size effect whereby larger populations develop more structured languages. Our results show that there are multiple ways to generalize from single agent pairs to larger populations, and that these design choices matter when it comes to studying the emergent language.

## 2 Communication Game

We study communication in referential games, a variant of the Lewis signaling game [37] proposed by Lazaridou et al. [34]. The game proceeds as follows: during each round, a sender agent $\pi$ observes an object $x \in \mathcal{X}$ (e.g., an arbitrary categorical entity, or a natural images) sampled from input space $\mathcal{X}$ according to distribution $p$ and generates a message $m \sim \pi(\cdot \mid x)$. Messages consist of variable length sequences of tokens picked from a discrete vocabulary $V$. Note that the tokens themselves are arbitrary and meaningless (typically they are represented as numbers from 1 to $|V|$). A receiver agent $\rho$ then observes message $m$ and must predict the original object from among a set of candidates $\mathcal{C} = \{x, y_1, \ldots y_{|\mathcal{C}-1|}\}$ containing $x$ and $|\mathcal{C}| - 1$ distractors, where each distractor $y$ is sampled uniformly without replacement from the input space excluding the original object, $\mathcal{X} \setminus \{x\}$. Concretely, this is implemented by calculating a score $f(y, m)$ for each candidate $y$ and defining the probability of a candidate conditioned on the message $\rho(\cdot \mid m, \mathcal{C})$ as $\frac{e^{f(x,m)}}{\sum_{y \in \mathcal{C}} f(y,m)}$. Based on the receiver's success, the sender agent receives a reward $R(x, \rho(\cdot \mid m, \mathcal{C}))$.

In practice, both senders and receivers are implemented as neural networks $\pi_\theta$ and $\rho_\psi$ with parameters $\theta$ and $\psi$ estimated by gradient descent. The sender is trained to maximize its expected reward using the REINFORCE algorithm [57], while the receiver maximizes the expected log-likelihood of identifying the original object, $\log \rho_\psi(x \mid m, \mathcal{C})$ (also known as the InfoNCE objective; Oord et al. [45]). Denoting as $\mathbb{E}_{x \sim p}$ the expectation over $x$ sampled from $p$, the corresponding training objectives are:

$$J_s(\theta) = \mathbb{E}_{x \sim p} \mathbb{E}_{m \sim \pi_\theta(\cdot|x)} \mathbb{E}_{\mathcal{C} \sim p} R(x, \rho_\psi(\cdot \mid m, \mathcal{C})) \tag{1}$$

$$J_r(\psi) = \mathbb{E}_{x \sim p} \mathbb{E}_{m \sim \pi_\theta(\cdot|x)} \mathbb{E}_{\mathcal{C} \sim p} \log \rho_\psi(x \mid m, \mathcal{C}) \tag{2}$$

### 2.1 Population Level Training

The two-player referential game can be generalized to larger populations of agents [41, 10]. In the most general case, we consider a population of $N_s$ senders and $N_r$ receivers that are linked by a bipartite *communication graph* $G$ defining connections between senders and receiver $(\pi_{\theta_i}, \rho_{\psi_j})$ [24, 30]. At training time, sender-receiver pairs are repeatedly sampled and trained to perform a round of the game. Importantly, only agent pairs that are connected in the communication graph are sampled. Throughout this paper, we will refer to this type of training as **Standard** training.

With this training procedure, agents are trained to maximize their communicative success with all their neighbors in the communication graph. Let $\mathcal{N}_G(i)$ refer to the neighbors of the $i$-th node in the graph, and $J_{s,i \rightarrow j}$ (respectively $J_{r,i \rightarrow j}$) denote the objective of $\pi_{\theta_i}$ (respectively $\rho_{\psi_j}$)) in the

pairwise communication from sender $i$ to receiver $j$. We can write the overall objective for sender $i$ (and receiver $j$, respectively) as:

$$J_{s,i}(\theta_i) = \frac{1}{|\mathcal{N}_G(i)|} \sum_{j \in \mathcal{N}_G(i)} J_{s,i \to j}(\theta_i) \quad \text{and} \quad J_{r,j}(\psi_j) = \frac{1}{|\mathcal{N}_G(j)|} \sum_{i \in \mathcal{N}_G(j)} J_{r,i \to j}(\psi_j). \quad (3)$$

At test time, the population is evaluated by averaging the referential accuracy across all possible sender-receiver pairings. Following previous work, in this paper we focus on populations with an equal number $N := N_s = N_r$ of senders and receivers, meaning that there are up to $N^2$ possible pairings.

## 2.2 What does Population-level Training Optimize?

To shed light on the differences between training a single agent pair and training a population of agents, we analyze the objective optimized by the population. Inspired by [1]'s analysis in the two-player case, we study the behaviour of the population at the optimum, that is when senders and receivers have reached a Nash equilibrium [46].

In this section, we make the simplifying assumption that $\mathcal{C} = \mathcal{X}$. In other words, receiver agents must pick the correct candidate out of all possible objects in $\mathcal{X}$. This allows us to remove the conditioning on $\mathcal{C}$ and write $\rho_\psi(x \mid m, \mathcal{C}) = \rho_\psi(x \mid m)$. We make this simplification to reduce clutter in notations. Nevertheless, our key observations still hold for $\mathcal{C} \neq \mathcal{X}$ (see Appendix B for a detailed discussion).

At a Nash equilibrium, the optimal receiver parameters $\psi_j^*$ satisfy

$$\rho_{\psi_j^*} = \arg\max_{\psi_j} J_{r,j}(\psi_j) = \arg\max_{\psi_j} \frac{1}{|\mathcal{N}_G(j)|} \sum_{i \in \mathcal{N}_G(j)} J_{r,i \to j}(\psi_j). \quad (4)$$

Assuming that receiver $\rho_{\psi_j}$ has high enough capacity, and training is able to reach the global optimum, the solution of the optimization problem in Equation 4 has an analytical solution $\rho_{\psi_j^*}$ which can be written as a function of $\pi_{\mathcal{N}_G(j)}^*(m \mid x) := \frac{1}{|\mathcal{N}_G(j)|} \sum_{i \in \mathcal{N}_G(j)} \pi_{\theta_i^*}(m \mid x)$, the mixture of all senders communicating with receiver $j$:

$$\rho_{\psi_j^*}(x \mid m) = \pi_{\mathcal{N}_G(j)}^*(x \mid m) = \frac{\pi_{\mathcal{N}_G(j)}^*(m \mid x) p(x)}{\mathbb{E}_{y \sim p} \pi_{\mathcal{N}_G(j)}^*(m \mid y)}.$$

In other words, $\rho_{\psi_j^*}$ is the posterior associated with $\pi_{\mathcal{N}_G(j)}^*$ (full derivation in appendix A).

An important implication of this result is that when the population graph is fully connected (all senders are connected to all receivers), each receiver converges to the same optimum $\pi^*(x \mid m) = \frac{\sum_{i=1}^n \pi_{\theta_i}(m \mid x) p(x)}{\mathbb{E}_{y \sim p} \sum_{i=1}^n \pi_{\theta_i}(m \mid x)}$, the posterior of the mixture of all senders in the population. Plugging this back into each sender's objective, we have

$$J_{s,i}(\theta_i^*) = \mathbb{E}_{x \sim p} \mathbb{E}_{m \sim \pi_{\theta_i^*}(\cdot \mid x)} R(x, \pi^*(\cdot \mid m))$$

Summing across all senders, we can rewrite the global objective optimized by the senders as

$$\max_{\theta^*} \mathbb{E}_{x \sim p} \mathbb{E}_{m \sim \pi^*} R(x, \pi^*(\cdot \mid m)). \quad (5)$$

In other words, at the equilibrium, the population maximizes the expected reward of the "sender ensemble" $\pi^*$, rather than that of individual agents $\pi_{\theta_i^*}$: the objective of a population $N$ agents is functionally the same irrespective of $N$. We postulate that this indifference to the population size may account for the surprising lack of effect of larger populations observed in some previous work [49, 10]. Differences in behaviour must be attributed to effects stemming from training dynamics (*e.g.* it becomes more difficult for receivers to learn the posterior $\pi^*(x \mid m)$), or be imposed through extraneous modifications of the population objective (for example explicit imitation components; Chaabouni et al. [10]).

A second observation is that there is no direct pressure for agents that communicate at training time to develop the same language. Indeed, it is entirely possible that all senders develop different but non-overlapping languages: it suffices that no two senders communicating with a shared receiver use the same message $m$ to describe a different object. In this case receivers can simply learn their neighboring sender's languages and there is no need for the senders to converge to a unified language.

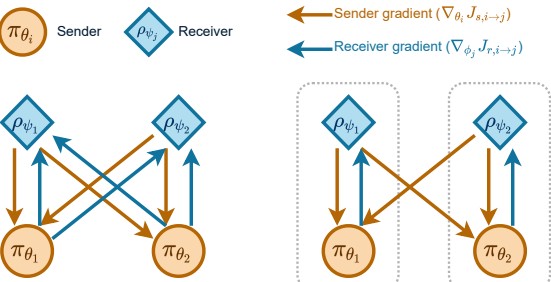

Figure 1: In the **standard** setting (left hand side), both receivers (in blue) are trained by maximizing their discrimination objective with respect to both senders. With **partitioning**, receiver $\rho_{\psi_1}$ (resp. $\rho_{\psi_2}$) is only trained to maximize its communication objective with sender $\pi_{\theta_1}$ (resp. $\pi_{\theta_2}$)

## 3 Partitioning Agents

A key difference between the usual population setting and populations of humans in laboratory experiments is that agents are not usually split into "senders" and "receivers". Rather, each participant in the experiment assumes both a sender and receiver role [21]. Our hypothesis is that, counter to what is customary in the emergent communication literature, tying senders and receivers is key in surfacing useful population-level dynamics in multi-agent communication.

To operationalize this sender-receiver coupling, we identify an "agent" as a sender-receiver pair. During training, we only train receiver $\rho_{\psi_i}$ with its associated sender $\pi_{\theta_i}$. In other words, $J_{r,i}(\psi_i) := J_{r,i \to i}(\psi_i)$. In doing so, we "partition" the agents by preventing receiver $i$ from co-adapting to other senders $j \neq i$. This procedure is illustrated in Figure 1. Note that senders can nevertheless still train with rewards from neighboring receivers, and so communication across agents can still emerge. Importantly, partitioning prevents receivers from learning to recognize multiple languages, as they are now only trained on messages emitted by a single sender.

Following a similar analysis as Section 2.2, we derive that at the optimum, receiver $\rho_{\psi_i^*}(x \mid m)$ now takes the form of the posterior associated with its respective sender, $\pi_{\theta_i^*}(x \mid m) = \frac{\pi_{\theta_i^*}(m|x)p(x)}{\mathbb{E}_{y \sim p} \pi_{m|y}}$ (derivation in Appendix A). We can thus write the population-level objective at the equilibrium as

$$\frac{1}{N} \sum_{i=1}^{N} \left[ \underbrace{\mathbb{E}_{x \sim p} \mathbb{E}_{m \sim \pi_{\theta_i^*}(\cdot|x)} R(x, \pi_{\theta_i^*}(\cdot \mid m))}_{\text{Internal communication}} + \underbrace{\sum_{j \in \mathcal{N}_G(i)} \mathbb{E}_{x \sim p} \mathbb{E}_{m \sim \pi_{\theta_i^*}(\cdot|x)} R(x, \pi_{\theta_j^*}(\cdot \mid m))}_{\text{Mutual intelligibility}} \right]. \quad (6)$$

Note that the functional form of the objective can now be decomposed into two parts: an "internal communication" objective which takes the same form as that of a single pair of agents, and a "mutual intelligibility" objective which enforces that neighboring agents are able to communicate successfully. In experiments, we show that this explicit pressure towards mutual intelligibility promotes the emergence of a single language within the population, which in turn enables agents to communicate with new partners outside of their training neighborhood.

## 4 Experimental Setting

### 4.1 Datasets

We perform experiments on two datasets: a simple, synthetic "attribute/values" dataset and a more realistic image dataset.

**Attribute/Values** In this dataset, each object is represent by a collection of abstract "attributes". Specifically, each input $x$ is a vector of 4 attributes, each of which can take 10 total values. This results in $10^4$ total attribute/value combinations [32, 9]. In each setting we hold out $1,000$ combinations to be used as a validation, and $1,000$ more for use as a test set. We can thus ensure that we are evaluating the agents' ability to generalize to unseen combinations of attributes.

**ImageNet** In addition to toy objects, we perform experiments with referential games based on more realistic objects. Following Chaabouni et al. [10], we use the ImageNet [17] dataset of natural images. The dataset consists of about 1.4M training images collected on the internet and annotated for 1,000 labels from the WordNet database [40]. Images are first encoded as 2048-sized real-valued vectors with a (frozen) ResNet pre-trained with BYOL [22] before being passed to sender and receivers.

## 4.2 Game Architecture

Both sender and receiver agents are based on 1 layer LSTMs [26] with embedding and hidden dimensions of size 256. Specifically, the sender first encodes the object $x$ into a vector of size 256, which is concatenated to the input of the LSTM. At each step, the output of the LSTM cell is passed through a fully connected layer to produce logits of size $|V|$. A softmax function is then applied to obtain normalized probabilities over the vocabulary. During training, messages are generated by sampling from the distribution whereas at test time we generate messages deterministically via greedy decoding. In both cases, generation stops whenever a special "<EOS>" is generated, or when the number of tokens reaches a fixed limit $L$.

The receiver encodes the message with an LSTM encoder, the output of which is the fed into a fully connected layer to yield a vector of size 512. The candidate objects $\mathcal{C}$ are then scored by computing the dot product of this vector with a 512-dimensional encoding of each candidate. The conditional distribution over candidates is then obtained by taking a softmax. We set the reward function for the sender to the log-likelihood assigned by the receiver to the correct candidate, $R(x, \rho_\psi(\cdot \mid m)) = \log \rho_\psi(x \mid m)$.

Throughout all experiments, we set the vocabulary size $|V|$ to 20 and the maximum length of the messages, $L$, to 10. This means that the communication channel used by the agents has a capacity of about $20^{10}$ which ensures that there is no communication bottleneck (the size of the channel is several orders of magnitude larger than the size of our datasets). Our implementation, based on the EGG toolkit [29], will be open-sourced upon de-anonymization.

## 4.3 Population training

We train populations following the procedure outlined by Chaabouni et al. [10]: for each minibatch of data, we sample $K$ pairs from the population (uniformly among the pairs linked in the communication graph). Each pair plays an episode of the game, and the agents are updated simultaneously following the gradients of their respective objectives. We take $K = \max(10, N)$ to ensure that each agent plays the game at least once at every step on average. This procedure needs to be modified for partitioned populations: since receiver $j$ is only with its respective sender instead of with all of its neighbors, there is now only a $\frac{1}{|N_G(j)|}$ chance that receiver $j$ will be updated every step (the probability that the pair $(j, j)$ is sampled). For

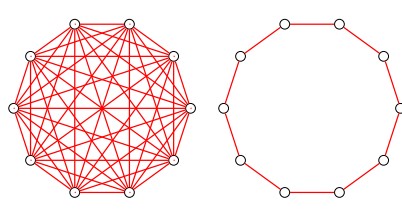

(a) Fully-connected     (b) Circular

Figure 2: Example of communication graphs used in this paper

larger populations, especially those that are fully-connected, this dramatically slows down training as receivers are updated very infrequently. To address this issue, we modify the procedure as follows: for every sampled agent pair $(\pi_{\theta_i}, \rho_{\psi_j})$, we calculate both $J_{s,i\rightarrow j}$ and $J_{r,i\rightarrow i}$ and update both $\pi_{\theta_i}$ and $\rho_{\psi_i}$. Note that this necessitates calculating both $\rho_{\psi_j}(x \mid m, \mathcal{C})$ and $\rho_{\psi_i}(x \mid m, \mathcal{C})$ and therefore we incur a small computational overhead. However we only observe a $\sim 5\%$ increase in training time due to the fact that we are back-propagating through only one of the two receivers, $\rho_{\psi_i}(x \mid m, \mathcal{C})$. With this modification, we recover the property that each agent (sender or receiver) is updated once every step on average.

In all experiments we train with a batch size of 1024 with the Adam optimizer [31] using a learning rate of 0.001 for the attribute/value dataset and 0.0001 for Imagenet. The other parameters are set to $\beta_1 = 0.9$, $\beta_2 = 0.999$ and $\varepsilon = 10^{-8}$. We apply $\ell_2$ regularization with a coefficient of $10^{-5}$.

We systematically augment the sender objectives with an entropy maximizing term, which has been found to encourage exploration [58]. The coefficient for this entropy term is set to 0.1 in all

Table 1: Accuracies with training partners and new partners on both datasets. Numbers are reported with standard deviation across all pairs for 3 independent experiments

|  | ImageNet | | Attribute/Values | |
|---|---|---|---|---|
|  | Standard | Partitioned | Standard | Partitioned |
| Training partners | 97.09 ±1.10 | 99.75 ± 0.08 | 99.88 ± 0.15 | 99.81 ± 0.22 |
| New partners | 5.41 ±13.57 | 96.24 ± 3.25 | 7.81 ± 18.28 | 40.37 ± 29.44 |

Table 2: Language similarity between training partners and new partners on both datasets. Numbers are reported with standard deviation across all pairs for 3 independent experiments

|  | ImageNet | | Attribute/Values | |
|---|---|---|---|---|
|  | Standard | Partitioned | Standard | Partitioned |
| Training partners | 0.28 ± 0.07 | 0.40 ± 0.02 | 0.28 ± 0.05 | 0.36 ± 0.01 |
| New partners | 0.22 ± 0.19 | 0.37 ± 0.15 | 0.23 ± 0.19 | 0.31 ± 0.17 |

experiments. To reduce the variance of the policy gradient in REINFORCE, we substract a baseline computed by taking the average reward within a given mini-batch for each pair [54].

We evaluate the population every epoch (every 5 epochs for the Attribute/Value dataset) on the validation set. We only evaluate on up to 100 unique pairs sampled uniformly within the population, this time without consideration for the communication graph. We train for a fixed number of epochs, selecting the best model based on the average validation accuracy across all evaluation pairs.

## 5 Communication with New Partners

In our first set of experiments, we evaluate the ability of agents trained in populations to communicate with partners they haven't interacted with during training.

### 5.1 Circular Populations

Specifically, we study "circular" populations of agents arranged on a ring lattice. Each agent (sender-receiver pair) $i$ is only trained with neighboring agents $i-1, \ldots, i+1$ and the graph is cyclical (see Figure 2b). We choose this type of population because it is an extreme case of a population where each agent has the same, minimal amount of neighbors (two), yet there is still a path between any two agents. In this context, *training partners* are sender-receiver pairs that are connected in the graph and have interacted during the training phase whereas *new partners* refers to pairs that have not interacted during training.

We report results along two metrics:

- **Communication Accuracy** of sender/receiver pairs on an evaluation set. This measures how successful the pair is in communicating.

- **Language Similarity** between senders. This metric (also called synchronization in Rita et al. [49]) is calculated as $1 - \delta_{i,j}$, where $\delta_{i,j}$ is the normalized edit distance between messages output by two senders, averaged across all objects in our evaluation set.

We report these metrics for both training partners and new partners. Note that high communication accuracy does not always entail similar languages: it is possible for the receivers to achieve high accuracy despite all senders sending different messages for any given object (it is only necessary for a given message to unambiguously refer to one object across senders).

### 5.2 Partitioning Enables Successful Zero-Shot Communication

In Table 1 and 2, we report accuracies and similarities for circular populations of 20 sender-receiver pairs trained on ImageNet and the Attribute/Values dataset. All metrics are calculated on the test set and averaged across 3 independent experiments.

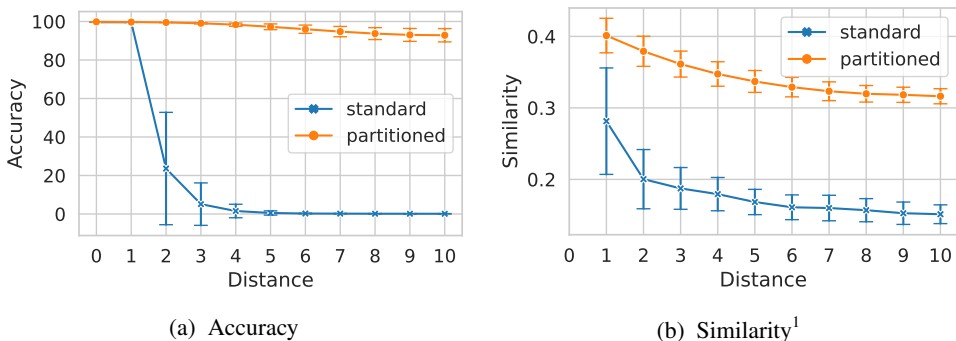

(a) Accuracy

(b) Similarity[1]

Figure 3: Accuracy and language similarity as a function of the distance between two agents in the communication graph.

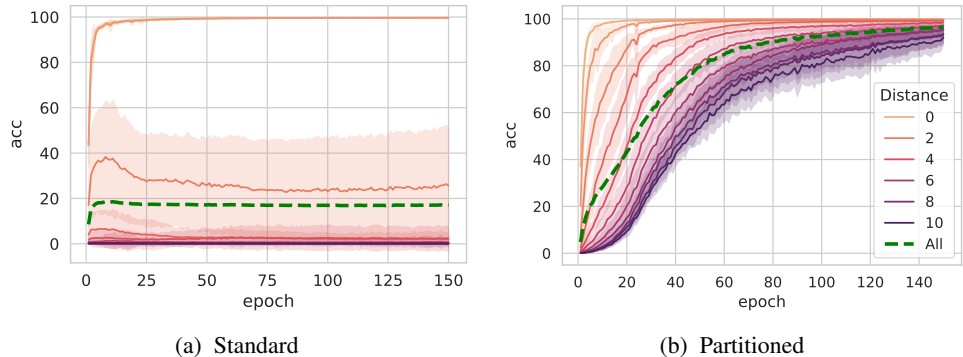

(a) Standard

(b) Partitioned

Figure 4: Evolution of validation accuracy during training across agent pairs at various distances in the communication graph. Results are aggregated over all agent pairs and 3 populations.

We observe that in populations following the standard training paradigm (**Standard**), there is a stark discrepancy between training and new partners. Indeed, on both datasets the accuracy with training partners reaches a very high value, above $95\%$. Yet, the accuracy when agents communicate with new partners drops down to less than $10\%$. On the other hand, in **Partitioned** populations, agents reach a much higher accuracy with non-neighbors, up to $96\%$ on ImageNet and $40\%$. A similar trend holds for language similarity.

Note that all metrics on new partners exhibit high standard deviation. An explanation is that among non-neighboring pairs there is a different behaviour depending on how far the two agents are in the population. This is verified in Figure 3, which displays a breakdown as a function of the distance between two agents in the communication graph (on ImageNet). We find that without partitioning, accuracy drops off sharply to close to 0 for agents at a distance $\geq 2$, whereas it decreases almost linearly with the distance in the partitioned case, down to about $95\%$ for the most distant agents.

### 5.3 Training dynamics

We further investigate the evolution of accuracies during training. In Figure 4, we plot the evaluation accuracies of both standard and partitioned populations broken down by distance between pairs, focusing on the ImageNet dataset. Note that there are two training phases in the standard case. Up to epoch $\approx 10$, the accuracy for all training pairs increases, after which agents *over-fit* to their training partners (distances 0 and 1) and the accuracy on other pairs decreases to a plateau.

On the other hand, Figure 4b illustrates the pressure for mutual-intelligibility in partitioned populations: as accuracy between training pairs reaches close to $99\%$ accuracy (around epoch 20), accuracies across distant pairs increases rapidly before plateauing above $90\%$. In fact, our results show that the most distant accuracies are still increasing after 150 epochs, albeit very slowly.

---

[1]By construction, the similarity of a sender with itself (corresponding to a distance of 0) is always one. We omit this value from the figure to better illustrate the trends for distance $\geq 1$.

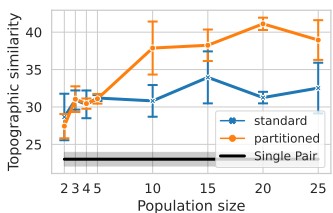 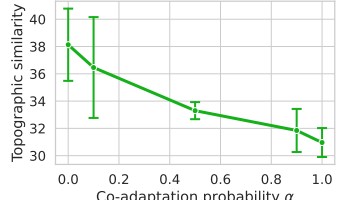 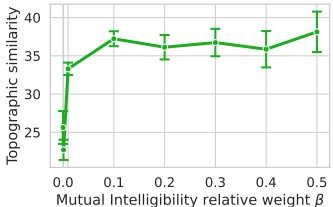

(a) Topographic similarity as a function of population size on an attribute/value communication game.

(b) Topographic similarity with varying degrees of partitioning (populations of size 10).

(c) Topographic similarity when ablating the mutual-intelligibility term (populations of size 10).

Figure 5: Influence of partitioning on the topographic similarity of the emergent languages.

## 6 Partitioned Population Develop More Compositional Languages

In this section, we investigate the effect of partitioning on the structure of the language, with a focus on *compositionality*.

### 6.1 Measuring Compositionality

A language is said to be compositional when the meaning of a whole utterance can be systematically deduced from the meaning of its components (*i.e.* words). The notion of compositionality is widely construed to underlay the near infinite productivity of human languages [55].

A common metric for measuring compositionality in emergent languages is the *topographic similarity* [5, 35]. Topographic similarity captures the intuition that a compositional language will map similar "meanings" to similar messages: the phrase "a red bird" is more similar to the phrase "a blue bird" than to "a powerful computer". In practice, the topographic similarity is computed by measuring the Spearman rank correlation coefficient [52] between (1) the pairwise distances across all objects and (2) the pairwise distance across all messages.

### 6.2 Effect of Population Size on Compositionality

We run experiments on our Attribute/Values dataset, with both standard and partitioned populations that are fully-connected (see Figure 2a). Population sizes range from 2 to 25 sender-receiver pairs. We compute topographic similarity using the Hamming distance in the object space (*i.e.* the distance between two objects is the number of attributes in which they differ) and the normalized edit distance between messages.

In Figure 5a, we observe that while standard population-level training does increase the topographic similarity of the language overall, population size has very little effect: populations of sizes 3 and 20 both reach about the same value of 30 on average. On the other hand, partitioning greatly increases the effect of population size on compositionality: populations of size 20 have a significantly higher topographic similarity than populations of size 5, with a $\approx 10$ points difference.

### 6.3 Co-adaptation is Responsible for the Decrease in Compositionality

Up until this point, we have described partitioning (or lack thereof) as a binary choice. However, it is possible to partition a population only partially, by allowing receiver $j$ to train with senders $i \neq j$ occasionally with probability $\alpha > 0$. In doing so, the optimal receiver now becomes the posterior associated with a mixture between $\pi_{\theta_i^*}(m \mid x)$ and $\pi^*(m \mid x)$ (see Appendix A for the derivation). If $0 < \alpha < 1$, receivers are now optimizing for a different objective (as in partitioned populations), but some amount of co-adaptation is still allowed.

We perform this experiment on the Attribute/Values dataset with a fully connected population of size 10, varying the degree of co-adaptation $\alpha$ ranging in $\{0, 0.1, 0.5, 0.9, 1\}$. $\alpha = 0$ corresponds to partitioned training whereas $\alpha = 1$ is equivalent to standard training. All populations converge to $> 99\%$ accuracy. However, in Figure 5b we find that topographic similarity drops as soon as we introduce minimal amounts of co-adaptation ($\alpha = 0.1$) and decreases steadily to the level of standard populations as $\alpha$ grows to 1. This further corroborates our hypothesis that reducing co-adaptation

promotes the emergence of a more structured language, and that eliminating it altogether (in a partitioned population) yields the best results.

## 6.4 Importance of Mutual Intelligibility

Recall that the objective of a partitioned population at the equilibrium (Equation 6) can be decomposed in two terms: an "internal communication" corresponding to the single agent pair objective and a "mutual intelligibility" term which encourages senders to align their languages. Importantly, the latter is the only element that separates a partitioned population from a collection of isolated agents.

To measure its effect on the compositionality of the emergent language, we train fully connected populations of size 10 and decrease the relative weight of the mutual intelligibility term. This is implemented by making the pair $(\pi_{\theta_i}, \rho_{\theta_i})$ more likely to be sampled than other pairs $(\pi_{\theta_i}, \rho_{\theta_j})$, $j \neq i$ by a factor $\times \frac{1-\beta}{\beta}$. We let $\beta$ range from $0.5$ (partitioned population) to $0.0$ (collection of isolated sender-receiver pairs). In Figure 5c, we find that emergent languages retain high topographic similarity even at small $\beta$, and the sharp drop-off occurs only when $\beta$ is very close to 0. This confirms that the mutual intelligibility term exerts a strong pressure towards compositionality. We investigate the evolution of the two terms during training in Appendix C.

## 7 Related Work

There is a rich history of modeling the emergence of language as the solution to a cooperative game that can be traced back to functional theories of language [59, 2, 13]. With a regain of interest for the study of language evolution [15, 12], a rich literature has developed around computational simulations of the emergence of language based on simple language games [37, 51, 3, 6]. Examples include studying evolutionary models of the emergence of grammar [44], the influence of cultural transmission [5], game theoretical considerations [27] or linguistic diversity [39] among others.

The recent success of deep learning in natural language processing has spurred interest in studying signaling games between deep neural network trained with reinforcement learning to solve a signaling game [34, 20]. Several follow-ups have taken this idea further by extending it to more complex games or environment [53, 25, 28, 16] or by adding an element of competition [50, 43] or negotiation [7] or even explicit pressure towards certain desirable properties [32, 11, 38, 48]. In parallel, several efforts have been made to understand the properties of the emergent languages [4, 8, 9].

Within this growing literature, multiple authors have explicitly studied the use of populations of more than two agents. Various works have argued for augmenting populations with an explicit pressure towards more structure languages, via *e.g.* generational transmission [14], adversarial regularization [56], varying learning speeds [49] or imitation learning and voting [10]. Although the focus is often on fully-connected populations, some authors have also explored more complex communication graphs, for the purpose of modeling contact linguistics [24] or the effect of social network structure on the language [19]. Recent work from Kim and Oh [30] is perhaps closest to our own: the authors study the effect of population size and connectivity in the standard training paradigm. In contrast, the purpose of this paper is to highlight the impact of the training procedure on these very effects.

## 8 Conclusion

Empirical findings in socio-linguistics suggest that population dynamics should help in simple sender-receiver communication games. In this paper, we observed that populations trained by naively extending the simple 1-1 protocol to $N \times N$ agent pairs fail to exhibit some of the properties that are observed in human populations. Motivated by an analysis of populations at the equilibrium, we described an alternative training paradigm, based on agents *partitioning* to reduce co-adaptation. Empirically, we find that partitioning enables us to recover some of the aforementioned properties.

Our findings call attention to the fact that there is more than one way to generalize two single to many agents, and simple design choices can have a great impact on the training dynamics and ultimately the effect of population on the emergent language. Beyond emergent communication, we hope that this observation can inspire similar work in other cooperative multi-agent problems where co-adaptation between agents may counteract population effects.

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

## A  Derivation of the Optimal Receiver

We first prove a more general result from which the optimal receiver both in the standard and partitioned can be derived.

### A.1  General Case

Consider a receiver $j$ trained to maximize

$$J_{r,j}(\psi_j) = \sum_{i \in \text{senders}} \alpha_i J_{r,i \to j}(\psi_j) \tag{7}$$

where $\alpha_{i=1\ldots n}$ are arbitrary weights for the senders (we assume that the $\alpha_i$ are positive and sum to one). We can rewrite the objective as:

$$
\begin{aligned}
J_{r,j}(\psi_j) &= \sum_{i \in \text{senders}} \alpha_i J_{r,i \to j}(\psi_j) \\
&= \sum_{i \in \text{senders}} \alpha_i \, \mathbb{E}_{m \sim \pi_{\theta_i}(\cdot | x)} \log \rho_{\psi_j}(x \mid m)
\end{aligned}
$$

Note that by linearity of expectation we can pass the $\alpha_i$ weighted average over the senders inside of the expectation and rewrite the second expectation in terms of the mixture $\pi_\alpha^*(m \mid x) := \sum_{i \in \text{senders}} \alpha_i \pi_{\theta_i^*}(m \mid x)$:

$$
\begin{aligned}
J_{r,j}(\psi_j) &= \mathbb{E}_{x \sim p} \, \mathbb{E}_{m \sim \sum_{i \in \text{senders}} \alpha_i \pi_{\theta_i^*}(m | x)} \log \rho_{\psi_j}(x \mid m) \\
&= \mathbb{E}_{x \sim p} \, \mathbb{E}_{m \sim \pi_\alpha^*(\cdot | x)} \log \rho_{\psi_j}(x \mid m)
\end{aligned}
$$

With slight abuse of notation, let us now denote by $\pi_\alpha^*(m) := \mathbb{E}_{x \sim p} \pi_\alpha^*(m \mid x)$ the marginal distribution over messages and $\pi_\alpha^*(x \mid m) := \frac{\pi_\alpha^*(m | x) p(x)}{\pi_\alpha^*(m)}$ the associated posterior. Notice that since by definition $\pi_\alpha^*(m \mid x) p(x) = \pi_\alpha^*(x \mid m) \pi_\alpha^*(m)$, we can rewrite the double expectation $\mathbb{E}_{x \sim p} \mathbb{E}_{m \sim \pi_\alpha^*(\cdot | x)}$ as $\mathbb{E}_{m \sim \pi_\alpha^*(\cdot)} \mathbb{E}_{x \sim \pi_\alpha^*(\cdot | m)}$ by inverting the order of summation. We can therefore rewrite

$$J_{r,j}(\psi_j) = \mathbb{E}_{m \sim \pi_\alpha^*(\cdot)} \, \mathbb{H}(\pi_\alpha^*(\cdot \mid m), \rho_{\psi_j}(\cdot \mid m))$$

where $\mathbb{H}(p, q)$ denotes the cross-entropy $\mathbb{E}_q[-\log p]$ of two distributions $p$ and $q$. Importantly the cross-entropy is non-negative and $\mathbb{H}(p, q) = 0$ if and only if $p = q$.

Consequently, the receiver $\rho_\psi$ will be optimal ($J_{r,j}(\psi_j) = 0$) if and only if for all $m$:[2]

$$\rho_{\psi_j^*}(x \mid m) = \pi_\alpha^*(x \mid m) = \frac{\pi_\alpha^*(m \mid x) p(x)}{\mathbb{E}_{y \sim p} \pi_\alpha^*(m \mid y)}. \tag{8}$$

$\square$

### A.2  Optimal Receiver in Standard Populations

Recall that in standard populations, the training objective for receiver $j$ is:

$$J_{r,j}(\psi_j) = \frac{1}{|\mathcal{N}_G(j)|} \sum_{i \in \mathcal{N}_G(j)} J_{r,i \to j}(\psi_j).$$

Note that this is a special case of Equation 7 with

$$
\alpha_i = \begin{cases} \frac{1}{|\mathcal{N}_G(j)|} & \text{if } i \in \mathcal{N}_G(j) \\ 0 & \text{otherwise} \end{cases}
$$

---

[2]More accurately, if the message space is not finite then the condition holds not for all $m$, but almost surely. However throughout the paper we are experimenting with finite (albeit large) message spaces.

Consequently, the derivation in Section A.1 tells us that the optimal receiver is

$$\rho_{\psi_j^*}(x \mid m) = \pi_{\mathcal{N}_G(j)}^*(x \mid m) = \frac{\pi_{\mathcal{N}_G(j)}^*(m \mid x)p(x)}{\mathbb{E}_{y \sim p} \, \pi_{\mathcal{N}_G(j)}^*(m \mid y)}. \tag{9}$$

Where $\pi_{\mathcal{N}_G(j)}^*(m \mid x) := \frac{1}{|\mathcal{N}_G(j)|} \sum_{i \in \mathcal{N}_G(j)} \pi_{\theta_i^*}(m \mid x)$

### A.3 Optimal Receiver in Partitioned Populations

In partitioned populations, the training objective for receiver $j$ is:
$$J_{r,j}(\psi_j) = J_{r,j \to j}(\psi_j).$$

This is also a special case of Equation 7 with
$$\alpha_i = \begin{cases} 1 & \text{if } i = j \\ 0 & \text{otherwise} \end{cases}$$

The derivation in Section A.1 thus yields the optimal receiver
$$\rho_{\psi_j^*}(x \mid m) = \pi_j^*(x \mid m) = \frac{\pi_j^*(m \mid x)p(x)}{\mathbb{E}_{y \sim p} \, \pi_j^*(m \mid y)}. \tag{10}$$

### A.4 Optimal Receiver in Partially Partitioned Populations

In the partially partitioned populations used in Section 6.3, each receiver's objective is a mixture between the standard and partitioned objective. This can also be rewritten as a special case of Equation 7 with
$$\alpha_i = \begin{cases} 1 - \alpha + \frac{\alpha}{|\mathcal{N}_G(j)|} & \text{if } i = j \\ \frac{\alpha}{|\mathcal{N}_G(j)|} & \text{if } i \in \mathcal{N}_G(j) \setminus \{i\} \\ 0 & \text{otherwise} \end{cases}$$

The optimal receiver can then be rewritten as the posterior distribution associated with the mixture sender
$$\alpha \times + (1 - \alpha) \times \pi_j^*(x \mid m)$$

## B   The Case of Referential Games

In the analysis from Section 2.2 onward, we assumed $\mathcal{C} = \mathcal{X}$ to simplify notation. We can relax this assumption without changing our key observation that all receivers are the same at the optimum.

Indeed, in this case the receiver's objective in a standard population is:

$$J_{r,j}(\psi_j) = \frac{1}{|\mathcal{N}_G(j)|} \sum_{i \in \mathcal{N}_G(j)} J_{r,i \to j}(\psi_j)$$

$$= \frac{1}{|\mathcal{N}_G(j)|} \sum_{i \in \mathcal{N}_G(j)} \mathbb{E}_{x \sim p} \, \mathbb{E}_{m \sim \pi_{\theta_i}(\cdot \mid x)} \, \mathbb{E}_{\mathcal{C} \sim p} \log \rho_{\psi_j}(x \mid m, \mathcal{C})$$

$$= \mathbb{E}_{x \sim p} \, \mathbb{E}_{m \sim \pi_{\mathcal{N}_G(j)}^*(\cdot \mid x)} \, \mathbb{E}_{\mathcal{C} \sim p} \log \rho_{\psi_j}(x \mid m, \mathcal{C})$$

This objective, called InfoNCE [45] also has an analytical solution that can be expressed as a function of $\pi_{\mathcal{N}_G(j)}^*$, of the form:

$$\rho_{\psi_j^*}(x \mid m, \mathcal{C}) = \frac{\frac{\pi_{\mathcal{N}_G(j)}^*(x \mid m)}{p(x)}}{\sum_{y \in \mathcal{C}} \frac{\pi_{\mathcal{N}_G(j)}^*(y \mid m)}{p(y)}} \tag{11}$$

Despite the more complicated form of the optimal receiver, the key ingredients to our analysis in Sections 2.2 and 3 are preserved: at the optimum, each receiver is a function of the posterior $\pi_{\mathcal{N}_G(j)}(x \mid m)$ associated with the communication partners to which it co-adapts. A similar analysis in partitioned populations shows that the optimum for receiver $j$ then only depends on the posterior associated with its respective sender $\pi_{\theta_j^*}$ instead.

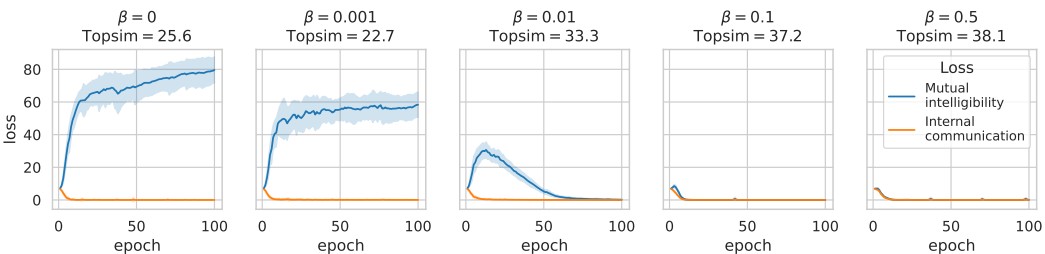

Figure 6: Evolution of internal communication and mutual intelligibility terms with different weightings $\beta$ (populations of size 10).

## C  Further Analysis of the Effect of Mutual Intelligibility

In Section 6.4, we find that languages stay highly compositional until the mutual intelligibility weight $\beta$ is decreased to almost 0. Our hypothesis is that even with small amounts of mutual intelligibility, agents will eventually have to optimize this part of the objective after they have maximized their respective internal communication to the point where the main contributor to the training gradient is the mutual intelligibility term.

To verify this hypothesis, in Figure 6 we report the evolution of both internal communication and mutual intelligibility losses during training for various values of the mutual intelligibility weight $\beta$. As expected, we observe that for all but very small values of $\beta$, the mutual intelligibility loss eventually decreases (although it decreases faster for high $\beta$).

