# OpenReview forum: "Revisiting Populations in Multi-Agent Communication"
_NeurIPS.cc/2022/Conference — NeurIPS 2022 Submitted_

### Official Review · Reviewer_z9wN · 2022-07-08

**Rating:** 6
**Confidence:** 3
**Soundness:** 3 good
**Presentation:** 2 fair
**Contribution:** 2 fair

**Summary:**

The research reevaluates the standard training procedure, highlights its shortcomings, and studied the communication of multi-agents at the population level.
The primary contributions: the paper propose an alternative training procedure that partitions sender-receiver pairs, restricts co-adaptation of receiver agents, and explicitly promotes mutual understanding between various agents. Moreover, the article discovers that languages created in partitioned populations are more compositional, and that population size has an effect, with bigger populations developing more structured languages.


**Questions:**

Can you make your code public?

**Limitations:**

Yes.

**Strengths And Weaknesses:**

Strengths:
In this study, multi-agent communication is examined from the viewpoint of the population. the idea is relatively creative, Moreover, population-level communication in sender-receiver Lewis games is analyzed, and a novel protocol is proposed, experiments also demonstrate the efficacy of the proposed protocol.
Weaknesses: The analysis of 6.3 may be unclear. It would not be able to understand it without reading the supplementary materials, but in 6.3 it does not say that supplementary materials are needed.

---

> ### Author Response · Authors · 2022-08-02
> **Response to Reviewer z9wN**
>
> We thank the reviewer for their encouraging comments!
>
> > The analysis of 6.3 may be unclear. It would not be able to understand it without reading the supplementary materials, but in 6.3 it does not say that supplementary materials are needed.
>
> We added an additional section in the appendix containing a more detailed derivation of the optimal receiver in the setting considered in setting 6.3. We have updated Section 6.3 to point to the appendix.
>
> > Can you make your code public
>
> We commit to making our code public upon de-anonymization. As mentioned in the paper, our code is based on EGG, an open source framework for emergent communication in pytorch.

---

### Official Review · Reviewer_cLW8 · 2022-07-11

**Rating:** 5
**Confidence:** 4
**Soundness:** 2 fair
**Presentation:** 3 good
**Contribution:** 2 fair

**Summary:**

This paper proposes a partitioned way of training communication in receiver-sender games in a network of receiver-sender agents. A theoretical analysis is carried and the optimality of popoulation training is discussed. The results show that a partitioned approach performs better when compared to a standard fully connected communication approach. In addiiton, the quality of the language learned by the agents is analysed.

**Questions:**

- according to line 246, the high standard deviations are due to the fact that different behaviour may show up between agents that are far away in the population. Does it mean that agents are then limited to communicate with their neighbours if they would be placed in a new setting? Based on the results with the distances between pairs (Fig 3) I would say that the distance does not seem to be a big problem.
- in section 6.4 and Figure 5c, the values remain stable until very low values of beta and only drop when mutual intelligibility is almost non-existent, raising questions regarding the importance of a lot of mutual intelligibility. It seems true that the existence of mutual intelligibility is needed for better compositionality, but why is there almost no difference from 0.1 to 0.5?
- in line 197, if for every sampled pair both objectives for sender and receiver i are calculated, from my understanding then it means that the neighbour senders of receiver i will also be updated (because it receives the message from the other receiver as in Fig 1) without updating their tied receiver. Will this have an impact on the language if many times only the sender of the pair will be updated, as it was initially mentioned to be a problem in line 190?

**Limitations:**

In the results, the compositionality demonstration could be improved; although the plots in Fig 5a give an idea of the argument, it would be interesting to visualise in some way the learned languages to a more clear perception of the compositionality.


Some minor mistakes/misspelings:
- in line 66, blank space at the end, seems that something is missing.
- line 294: “Figure 5c” -> “Figure 5b”
- line 79: “an” -> “a”
- line 94: “analyses” -> “analysis”.
- line 65: C is defined as a set of length |C| including x; thus “containing x and |C| distractors” is incorrect since the number of distractors should be |C|-1.
- line 192: “neighbor” -> “neighbors”



**Strengths And Weaknesses:**

Despite some grammar mistakes/misspelings (I document some below) this paper reads well.

Strengths:
-
The paper analyses theoretically a sender receiver system where there is exchange of messages and builds up to achieve the proposed partitioned setup. In addition, an interesting analysis of components such as language similarity/topographic similarity is carried.

Weaknesses:
-
- in line 74: “sampled from p”; p should be a distribution (or the input space X in this case); according to line 61, p is a probability value. Same problem in most equations throughout the paper, when writing for example “$E_{x\sim p}$”.
- in the objective of the sender in equation before line 111, why is $\pi$ conditioned on m in the second expectation $E_{m\sim\pi_{\theta_i^*(.|m)}}$? In all the previous definitions $\pi_i$ is always conditioned in the input, i.e., x.
- in line 138, the same derivation from section 2.2 is assumed to be valid to the partitioned setting. While I understand the logic for the standard setting (fully connected) I have concerns about whether the derivations still hold for the partitioned setting. I believe this requires some deeper explanation instead of simply stating it. For instance, in section 2 the objective of the sender is given as the average of the objectives of the neighbour senders of the sender i. But according to Fig 1, in the partitioned setting only the receivers have neighbour senders since the senders are only connected to receivers. Same question for receivers, since the neighbours of receivers are all senders according to Fig 1. I believe this should be made more clear.
- in line 104, from my understanding it is stated that the policy of the receiver is equal to the average policy of all the neighbour senders. It is unclear to me how this equality holds even when the receiver has enough capacity. At the optimal level I agree that the policy of the receiver could indeed be some mix of the policies of the neighbour senders but it is unclear to me why this mix is the average. Is this an assumption?

Generally, I have some concerns that I have outlined in this section and some questions below.

---

> ### Author Response · Authors · 2022-08-02
> **Response to Reviewer cLW8 (part 1)**
>
> We thank the reviewer for their comprehensive review and their attention to details.
>
> We believe that the reviewer's main concerns derive from a lack of clarity in our presentation, especially in Section 2, which we’ll attempt to clear up in this response and in our revision. We would be grateful if the reviewer could confirm whether our response addresses their issues with the paper and let us know if there are any outstanding concerns.
>
> > “sampled from p”; p should be a distribution
>
> The reviewer is correct that this results from poor phrasing on our part. Indeed, $p$ refers to a distribution and not a probability value: line 61 should read “sampled from input space X **according to distribution** p”. This was fixed in the revision.
>
> > in the objective of the sender in equation before line 111, why is π conditioned on m in the second expectation
>
> This is indeed a typo. In this equation, $\pi$ should be conditioned on $x$. This was fixed in the revision
>
> > In line 104 [...] it is stated that the policy of the receiver is equal to the average policy of all the neighbour senders. It is unclear to me how this equality holds even when the receiver has enough capacity.
>
> First, note that the optimal receiver is not the average of the sender policy, but rather the **posterior** of the average sender policy (as per the Eq. after line 105).
>
> That being said, this equality does hold at the optimum, when we maximize the objective with respect to the receiver. A full derivation can be found in appendix A. Informally, the reason for why the optimal receiver is corresponds to the (posterior of the) **average** of the neighboring speakers (and not any other weighting) is that the objective of the receiver (cf Eq. 3, right hand side) is the **average** of its pairwise communication objectives with all speakers.
>
> More generally, we can adapt the proof in appendix A to obtain a more general result for arbitrary weightings. Let us change the objective of the receiver $J^r_{i}$ to $\sum_{j\in \text{senders}}\alpha_jJ^r_{i\rightarrow j}$, where $\alpha_j$ are arbitrary weights (non-negative, summing to one). We can then show, following a similar proof as appendix A, that the optimal receiver in this case is the posterior associated with $\sum_{j\in \text{senders}}\alpha_j \pi_{{\theta^*}i}(m\mid x)$ , the $\alpha_i$ weighted sum over all senders. Our derivation in the paper corresponds to the special case where the weights are uniform over the neighborhood $N_G(i)$, $\forall j\in N_G(i), \alpha_j=\frac{1}{|N_G(i)|}$. Note that the partitioned setting is another special case, where weight is $1$ for the associated sender $\pi_{\theta_i}$ and $0$ elsewhere.  In our revision, we replaced the derivation in Appendix A with this more general case
>
> > in line 138, the same derivation from section 2.2 is assumed to be valid to the partitioned setting. While I understand the logic for the standard setting (fully connected) I have concerns about whether the derivations still hold for the partitioned setting. For instance, in section 2 the objective of the sender is given as the average of the objectives of the neighbour senders of the sender i. But according to Fig 1, in the partitioned setting only the receivers have neighbour senders since the senders are only connected to receivers. Same question for receivers, since the neighbours of receivers are all senders according to Fig 1. I believe this should be made more clear.
>
> To clarify the setting, communication always proceeds from sender to receiver. Consequently, the communication graph G is bipartite (as mentioned in Section 2.1), meaning that the neighbours of a sender are always receivers and vice-versa, both in the standard and partitioned setting. We are using the term “fully connected” with a slight abuse of notation, because we are referring to the case where all senders are connected to all receivers (but senders are not connected to senders, nor receivers to other receivers).
>
> As a consequence of this, in Eq. 3, the sender’s objective is the sum of its communication objectives with the neighbouring **receivers**, not senders. Similarly, in both cases (standard and partitioned), receivers are only connected to their neighbouring **senders**.
>
> This is why the optimal receiver can be expressed as a function of (the posterior of) the average policy of its neighbouring senders. In the partitioned setting, effectively we are only allowing the receivers to train with their respective senders. Concretely, this can be thought of as replacing the neighbouring senders $N_G(j)$ of receiver $j$ with a singleton $\{\pi_{\theta_j}\}$. We can thus reiterate the argument from line 104, replacing the average of senders in $N_G(j)$ with simply $\{\pi_{\theta_j}\}$.
>
> We hope that this clarifies the argument, but we are happy to discuss further. In our revision, we modified appendix A to contain an explicit derivation of the optimal listener in the partitioned case as well.
>
> [see next comment]

---

> > ### Author Response · Authors · 2022-08-02
> > **Response to Reviewer cLW8 (part 2)**
> >
> > [continuation of our response]
> >
> > > according to line 246, the high standard deviations are due to the fact that different behaviour may show up between agents that are far away in the population. Does it mean that agents are then limited to communicate with their neighbours if they would be placed in a new setting? Based on the results with the distances between pairs (Fig 3) I would say that the distance does not seem to be a big problem.
> >
> > Indeed, in the circular populations considered in this section, agents only communicate with their immediate neighbours at training time, but they are evaluated against distant neighbours with which they have not been trained.
> >
> > As the reviewer correctly points out, in Fig.3 there is a relatively small difference between close and distant pairs (in the partitioned setting), and indeed this is reflected in the standard deviations in Table 1: overall deviation is lower for the partitioned setting (~= 3) than for the standard setting where there (>10) where there is a large difference between close and distant pairs (as per Figure 3, orange lines)
> >
> > > in section 6.4 and Figure 5c, the values remain stable until very low values of beta and only drop when mutual intelligibility is almost non-existent, raising questions regarding the importance of a lot of mutual intelligibility. It seems true that the existence of mutual intelligibility is needed for better compositionality, but why is there almost no difference from 0.1 to 0.5?
> >
> > The reviewer touches an interesting point. We believe that this is a result of the fact that we are looking at performance after convergence. Our hypothesis is that even with small amounts of mutual intelligibility, agents will eventually have to optimize this part of the objective after they have maximized their respective internal communication to the point where the main contributor to the training gradient is the mutual intelligibility term. In other words, there might be little difference between 0.1 and 0.5 because in both cases if the agents want to decrease the mutual intelligibility part they have to develop a more compositional language.
> >
> > We include an additional figure confirming this hypothesis in Appendix C, Fig. 6. The figure shows the evolution of both internal communication and mutual intelligibility losses during training for various values of the mutual intelligibility weight $\beta$. As expected, we observe that for all but very small values of $\beta$, the mutual intelligibility loss eventually decreases (although it decreases faster for high $\beta$). Section 6.4 was updated to point to this additional analysis.
> >
> > > in line 197, if for every sampled pair both objectives for sender and receiver i are calculated, from my understanding then it means that the neighbour senders of receiver i will also be updated (because it receives the message from the other receiver as in Fig 1) without updating their tied receiver. Will this have an impact on the language if many times only the sender of the pair will be updated, as it was initially mentioned to be a problem in line 190?
> >
> > As we have hopefully clarified earlier in our response, in our communication game messages systematically go from sender to receiver. In the partitioned setting, an episode between a sender $i$ and a receiver $j$ proceeds as follows:
> >
> > 1. Sender $i$ sends a message to receiver $j$, but also receiver $i$
> > 2. Sender $i$ receives reward from receiver $i$ and $j$ and uses it to update its parameters
> > 3. Receiver $i$ calculates its reconstruction/discrimination objective based on sender $i$’s message and uses it to update its parameters
> >
> > Note that receiver $j$ is not updated. Moreover sender $j$ does not intervene in this procedure, and so it is not updated either. Only sender $i$ and receiver $i$ are updated. Consequently there is a parity of updates between senders and receivers. If the reviewer thinks this would be useful, we could include this more detailed explanation in the camera ready version
> >
> > > Some minor mistakes/misspellings
> >
> > We thank the reviewer for their attention to detail. We have addressed these typos/oversights in the revision.

---

> > > ### Comment · Reviewer_cLW8 · 2022-08-09
> > > **Response to authors response**
> > >
> > > Thanks for replying to my comments. I believe most of my concerns were addressed after the discussion. I think the paper is more clear now after the revisions made by the authors.
> > >
> > > ---
> > > Stated in the authors response:
> > > > Note that receiver  is not updated. Moreover sender  does not intervene in this procedure, [...] If the reviewer thinks this would be useful, we could include this more detailed explanation in the camera ready version.
> > >
> > > I do think this could be better explained later since it is not explicit in the paper and it can create some confusion.
> > >
> > > As a side note, although this is understandable in the context, I would also give some thought to the notation used in the new Appendix A.4 when using $\alpha$ now with 2 different meanings (weights and probability) as it can be confusing.
> > >
> > > Along with the lines of what mine and other reviews have pointed, I still think this paper would benefit from a better understanding of the languages learned and their structures, for example by visualizing them in a better way.
> > >
> > > Nonetheless, the paper is interesting and has improved with the revisions and it is more clear now. With all this being said, I will change my score.

---

### Official Review · Reviewer_sLjb · 2022-07-12

**Rating:** 4
**Confidence:** 4
**Soundness:** 4 excellent
**Presentation:** 4 excellent
**Contribution:** 3 good

**Summary:**

The authors propose a modified training procedure for population-based training for emergent communication in which specific speaker-listener pairs are constructed, and replace the "free for all" mixing of pairs present in traditional methods.  Constraining the mixing in this way limits co-adaptation, and thus improves generalization and topographical similarity.  In addition, this solution also solves some more long-standing issues, like why scaling to larger populations in traditional training methods has not typically met with significant improvements in emergent structure or task accuracy, and the authors demonstrate how this originates in the loss of the standard training algorithm.


**Questions:**

Questions are mostly implied by 'weaknesses', but to be clear: how does your proposed method relate to neural iterated learning, cultural transmission, and ease-of-teaching based approaches which study similar dynamics and reach similar conclusions -- typically higher task accuracy and topographic similarity?

**Limitations:**

I did not see any explicit discussion of limitations or societal impact, but it is hard to imagine such small scale experiments in this topic having important negative societal impact.

**Strengths And Weaknesses:**

Strengths:

Overall I really enjoyed this paper.  It's very well written, and the way the authors narrow their focus to this one specific issue helps the argument structure come across very clearly.  I thought their argument was convincing, especially when the scope and argument applies to training dynamics in toy scenarios.  The paper contributes important theoretical understanding to population-based optimization for EC, and the effects of controlling population size, a co-adaptation "factor", and mutual intelligibilty.

Weaknesses:
I don't have any major criticism of the work within the scope the authors layout.  However, working against this paper is that the authors make only a passing effort to connect their work to previous related work, some of which may have very similar interpretations.  Two strands of work that I think should be discussed in much more detail and direct comparison are are neural iterated learning and cultural transmission models.  Both approaches are mentioned in the related work, but no direct comparison is made, despite that both approaches apply similar constraints to the population, essentially limiting who is talking to who, and likely reducing co-adaptation.  Considering the improved performance on communicating with novel partners, one would think some comparison to iterated learning, or the Easy-of-teaching (Li 2019) would have been warranted here.

I think by the phrasing of the last paragraph of the related work (L332/333) the authors are trying to position themselves outside of this work, but I don't find it a convincing excuse.  The motivations are similar, the solutions are quite similar, and it would be appropriate to connect to this research, both in discussion and in experimentation.  Ideally I would like to see existing work as baselines in direct comparison to this work, exploring the role of different partitioning patterns have.

Because in the end, I'm unsure of what these results would mean in terms of pushing the field forward.  If iterated learning is more effective on all of the proposed benchmarks, is this just a footnote in the progression of EC research or only relevant to those who are for some reason unwilling to consider known approaches to training that work better than the standard population-based training scheme?  If the authors more explicitly compared their partitioning approach to other population-based approaches, it would be clearer what the impact of the research would mean in a more practical sense.

Intuitively, to me, the other approaches seem more a bit more plausible from a sociolinguistic point-of-view.  Of course there is no need for the partitionings to exist in this circular pattern, but I'm not sure how these results relate to existing work when more realistic communication cliques are used.  Some discussion of partitioning as it relates to a plausible force in shaping human language development would be appreciated.

My review score reflects that I believe this paper has a lot of merit, but comes up short in connecting to existing research and acknowledging extent and similarities of those contributions to a sufficient extent.

---

> ### Author Response · Authors · 2022-08-02
> **Response to Reviewer sLjb**
>
> We thank the reviewer for their in depth feedback. As we understand it, the reviewer’s key concern with the paper is its relationship with existing work, especially the iterated-learning/ease of learning line of research. Below, we address specific points separately. We hope this addresses the reviewer’s concerns, and we are happy to discuss more.
>
> > The paper is not well connected with existing research on iterated learning (& affiliated literature)
>
> We agree with the reviewer that the relationship between the population setting considered in our work and work on generational transmission could be made more explicit. We propose to incorporate a dedicated paragraph in the related work clarifying the distinction between our setting and the generational transmission setting.
>
> *There is a longstanding line of research in the language evolution focused on iterated learning [1, 2, 3], which posits that the structure of natural languages is shaped by the competing pressure of effective communication (expressivity) and the bottleneck of generational transmission (learnability). These theories have been applied in the neural emergent communication literature in various forms, such as neural iterated learning [4,5] or ease-of-teaching [6]. In these approaches, generally the goal is to train a population of agents spread out “temporally” across multiple generations. For example, in Ren et al. (2020) [3] there are only two agents interacting in each generation, and subsequent generations are connected via cultural transmission (the next generation imitates the previous one).*
>
> *On the other hand, our work focuses on modelling populations of agents spread-out “spatially” and interacting during a single generation. Indeed, there is documented evidence in the cognitive science literature (see e.g. [7, 8]) that population size or topology can influence properties of emergent languages in human experiments, even in a single generation (i.e. without a generational transmission bottleneck). Moreover, this enables us to study phenomena proper to large populations of interacting agents such as multilinguality.*
>
> Exploration of the interaction between the effects of “spatial” or “horizontal” spread (as in this paper) and “temporal” spread (ie. generational transmission) in populations would be warranted, but falls outside the scope of this paper.
>
> > The scope of this paper is too limited
>
> We would like to argue that there is significant interest recently in studying language emergence within populations *without* generational transmission, see e.g.  [9, 10, 11, 12]. We believe that uncovering the mechanisms underlying the effects of population-level training, and pointing out inadequate modelling decisions (standard vs. partitioned) is a useful contribution to the literature.
>
> > Some discussion of partitioning as it relates to a plausible force in shaping human language development would be appreciated.
>
> Our inspiration for partitioning was that “agents” (humans) in communication experiments generally assume the role of both sender and receiver, and both speaking and listening functions are inherently tied [13]. Although this motivation was hinted at in Section 3, we propose to make it clearer by adding the following sentence to the introduction:
>
> *As our main contribution, we propose an alternative training procedure which \emph{partitions} sender-receiver pairs and limits co-adaptation of receiver agents. **In partitioned populations, agents can be identified with sender-receiver pairs where both functions (sending and receiving) are inherently tied and cross-agent co-adaptation is limited.**  We show that this new training paradigm maximizes a different objective at the population level. In particular, it explicitly promotes mutual-intelligibility across different agents.*
>
> [see next comment for bibliography]

---

> > ### Author Response · Authors · 2022-08-02
> > **Response to Reviewer sLjb (bibliography)**
> >
> >
> > ## References
> > - [1]: Kirby, Simon, and James R. Hurford. "The emergence of linguistic structure: An overview of the iterated learning model." Simulating the evolution of language (2002): 121-147.
> > - [2]: Kirby, Simon, Hannah Cornish, and Kenny Smith. "Cumulative cultural evolution in the laboratory: An experimental approach to the origins of structure in human language." Proceedings of the National Academy of Sciences 105.31 (2008): 10681-10686.
> > - [3]: Beckner, Clay, Janet B. Pierrehumbert, and Jennifer Hay. "The emergence of linguistic structure in an online iterated learning task." Journal of Language Evolution 2.2 (2017): 160-176.
> > - [4]: Ren, Yi, et al. "Compositional languages emerge in a neural iterated learning model." ICLR. 2019.
> > - [5]: Lu, Yuchen, et al. "Countering language drift with seeded iterated learning." International Conference on Machine Learning. PMLR, 2020.
> > - [6]: Li, Fushan, and Michael Bowling. "Ease-of-teaching and language structure from emergent communication." NeurIPS 2019.
> > - [7]: Raviv, Limor, Antje Meyer, and Shiri Lev-Ari. "Larger communities create more systematic languages." Proceedings of the Royal Society B 286.1907 (2019)
> > - [8]: Raviv, Limor, Antje Meyer, and Shiri Lev-Ari. "Compositional structure can emerge without generational transmission." Cognition 182 (2019)
> > - [9]: Graesser, Laura Harding, Kyunghyun Cho, and Douwe Kiela. "Emergent Linguistic Phenomena in Multi-Agent Communication Games." EMNLP. 2019.
> > - [10]: Kim, Jooyeon, and Alice Oh. "Emergent communication under varying sizes and connectivities." NeurIPS (2021):
> > - [11]: Chaabouni, Rahma, et al. "Emergent communication at scale." ICLR 2022.
> > - [12]: Rita, Mathieu, et al. "On the role of population heterogeneity in emergent communication." ICLR 2022.
> > - [13] Hockett, Charles F., and Charles D. Hockett. "The origin of speech." Scientific American 203.3 (1960): 88-97.

---

> > > ### Comment · Reviewer_sLjb · 2022-08-08
> > > **Looking for empirical comparisons to existing work**
> > >
> > > Thank you to the authors for replying to my comments.
> > >
> > > I like the addition to the related work, as it gives a good high-level view comparing the two strands of work.  However my criticism is really a call for a more substantive empirical comparison to previous work.  In short, I'm sure there are many ways to restrict and constrain communication dynamics -- this paper proposes one, of a certain "pattern" within a population, and neural iterated learning proposes one across generations.  A natural other would be any of the possible "patterns" possible in the authors' proposed method not empirically studied, and I'm sure there are more that we're not even considering.
> > >
> > > When proposing a new constraint on communication dynamics, it seems important, if not necessary, to directly experimentally compare with that work to ensure that multiple constraint implementations are not exerting essentially the same pressures.  A paraphrase of something I mentioned in my initial review, if the success of the agents and the structure of the induced languages are comparable, what would we learn from this particular work?
> > >
> > > I believe knowing the extent to which these pressures overlap with those created by the methods in existing work could significantly affect the importance of the work.
> > >
> > > The consequence of not performing these experiments is that it opens the door for submissions of any permutation of how agents might be constrained in their communication patterns.  While this paper *is* a good paper, not performing these sorts of due-diligence types of experiments to understand empirically the proposed methods with respect to existing work means that the paper does not seem to me to meet the requirements for recommendation at a top-tier venue.
> > >
> > > A less important point IMO, but when I was referring to relating this work to a plausible explanation of how these pressure arise naturally, I meant more along the lines of -exactly- the approach in this work, using this "pattern" where each agent talks to one other, forming this very artificial graph structure.  I would have liked to see the effect of various patterns, and maybe ones which have a more plausible natural structure.  Would co-adaptation increase quickly as the structure degrades?  There were experiments that were maybe spiritually along these lines, but did not actually shake up the communication graph structure.

---

### Author Response · Authors · 2022-08-02
**General Response**

We thank the reviewers for their comments. We respond to their respective concerns in dedicated comments. From our understanding the general issues with the paper went along two lines:

1. Connection with the existing literature, especially on generational transmission (Reviewer sLjb)
2. Lack of clarity (Reviewers cLW8 and z9wN)

We have uploaded an updated version of our draft including several changes addressing these issues. We list the most significant changes below:

- Reworked appendix A to provide derivation for optimal receivers in all the populations considered (standard, partitioned, partially partitioned), and linked to the derivations in the relevant places (Section 2, 3, 6.3) [Reviewers cLW8 and z9wN]
- Added a more in depth discussion of the results of Section 6.4 in Appendix C, including an additional figure to explain why the dropoff in compositionality only occurs at very low mutual intelligibility weight [Reviewer cLW8]
- Fixed a number of typos and inaccuracies [mostly pointed out by Reviewer cLW8]

The following additional changes could not be added to the current revision due to space constraints, but will be added in the camera ready version:

- Added a paragraph in the related work dedicated to contextualising and contrasting our work with the literature on emergent communication with generational/cultural transmission. [Reviewer sLjb]
    - *There is a longstanding line of research in the language evolution focused on iterated learning [1, 2, 3], which posits that the structure of natural languages is shaped by the competing pressure of effective communication (expressivity) and the bottleneck of generational transmission (learnability). These theories have been applied in the neural emergent communication literature in various forms, such as neural iterated learning [4,5] or ease-of-teaching [6]. In these approaches, generally the goal is to train a population of agents spread out “temporally” across multiple generations. For example, in Ren et al. (2020) [3] there are only two agents interacting in each generation, and subsequent generations are connected via cultural transmission (the next generation imitates the previous one). On the other hand, our work focuses on modelling populations of agents spread-out “spatially” and interacting during a single generation. Indeed, there is documented evidence in the cognitive science literature (see e.g. [7, 8]) that population size or topology can influence properties of emergent languages in human experiments, even in a single generation (i.e. without a generational transmission bottleneck). Moreover, this enables us to study phenomena proper to large populations of interacting agents such as multilinguality.*
- Made our motivation for partitioning as tying sender and receiver more explicit in the introduction [Reviewer sLjb]
    - *As our main contribution, we propose an alternative training procedure which \emph{partitions} sender-receiver pairs and limits co-adaptation of receiver agents. **In partitioned populations, agents can be identified with sender-receiver pairs where both functions (sending and receiving) are inherently tied and cross-agent co-adaptation is limited.**  We show that this new training paradigm maximizes a different objective at the population level. In particular, it explicitly promotes mutual-intelligibility across different agents.*

---

### Author Response · Authors · 2022-08-08
**Reviewer discussion**

We appreciate the reviewers' time and effort in providing feedback on our submission.

As the author-reviewer discussion period draws to a close, we look forward to hear whether our response addressed their concerns and are happy to engage in further discussion if there are still outstanding issues with the paper.

Best,

---

### Meta-Review · Area_Chair_dBr1 · 2022-08-26

**Recommendation:** Reject
**Confidence:** Certain

**Metareview:**

The paper investigates the effectiveness of population-level training of multi-agent communication strategies. Based on the finding that agents that interact with one another co-adapt, the paper proposes an alternative training process that "partitions" the population by constructing specific sender-receiver pairs in a manner that reduces co-adaptation. The paper shoes that this partition-based strategy gives rise to a new optimization objective that encourages alignment across the population. Experiments demonstrate the emergence of mutual understanding between agents that have never communicated, and that partition-based training results in language that is more compositional compared to alternative strategies.

The paper was reviewed by three researchers who discussed the merits of the paper with the AC. There is general agreement that the paper provides an interesting discussion of and important insights into the effect of population-based optimization for emergent communication. Based on these insights, the authors propose a novel training procedure that experiments show is effective. Several reviewers commented that the paper is very well written and was enjoyable to read. The reviewers raised several concerns/questions that the authors made a concerted effort to address. However, a notable limitation of the current version of the paper is the lack of qualitative and quantitative comparisons to previous work. While the proposal to update the related work discussion is helpful, the paper should also provide experimental evaluations that compare to existing work, without which the significance of this particular training procedure is unclear.

**Award:**

No

---

### Decision · Program_Chairs · 2022-09-14

Reject